# AdaVAE: Bayesian Structural Adaptation for Variational Autoencoders

**Paribesh Regmi**     **Rui Li**[*]
Rochester Institute of Technology
`{pr8537, rxlics}@rit.edu`

## Abstract

The neural network structures of generative models and their corresponding infenrence models paired in variational autoencoders (VAEs) play a critical role in the models' generative performance. However, powerful VAE network structures are hand-crafted and fixed prior to training, resulting in a one-size-fits-all approach that requires heavy computation to tune for given data. Moreover, existing VAE regularization methods largely overlook the importance of network structures and fail to prevent overfitting in deep VAE models with cascades of hidden layers. To address these issues, we propose a Bayesian inference framework that automatically adapts VAE network structures to data and prevent overfitting as they grow deeper. We model the number of hidden layers with a beta process to infer the most plausible encoding/decoding network depths warranted by data and perform layer-wise dropout regularization with a conjugate Bernoulli process. We develop a scalable estimator that performs joint inference on both VAE network structures and latent variables. Our experiments show that the inference framework effectively prevents overfitting in both shallow and deep VAE models, yielding state-of-the-art performance. We demonstrate that our framework is compatible with different types of VAE backbone networks and can be applied to various VAE variants, further improving their performance.

## 1   Introduction

The inference models and the generative models paired in variational autoencoders (VAEs) are commonly constructed with neural networks, i.e., encoding networks and decoding networks, respectively [1, 2, 3]. Extensive research efforts show that well-designed encoding/decoding network structures for VAE models can constantly achieve state-of-the-art generative performance compared to other generative models [4, 5, 6]. However, powerful VAE network structures are hand-crafted and fixed prior to training. The issue with fixed network structures is that shallow ones limit VAE models' expressiveness, whereas overly deep networks are slow to use and prone to overfitting. Traditional model selection by training different VAE network structures for given data is difficult since finding optimal hyperparameters for each candidate structure is a daunting task, and training large VAE structures requires significant computation. On the other hand, network structure adaptation methods for discriminative model settings [7, 8, 9] cannot be straightforwardly applied to address the unique challenge posed by VAE estimation along with the latent variables.

Although the network structures play a critical role in the performance of VAE models, they are largely overlooked by current VAE regularization methods. This renders their failure to prevent overfitting when the network structures grow deeper. Amortized inference regularization (AIR) proposes two approaches: injecting random noise to the VAE objective for inference or directly restricting the inference models to a set of smooth functions [10, 11]. Another approach in VAE regularization

---

[*]Corresponding author

37th Conference on Neural Information Processing Systems (NeurIPS 2023).

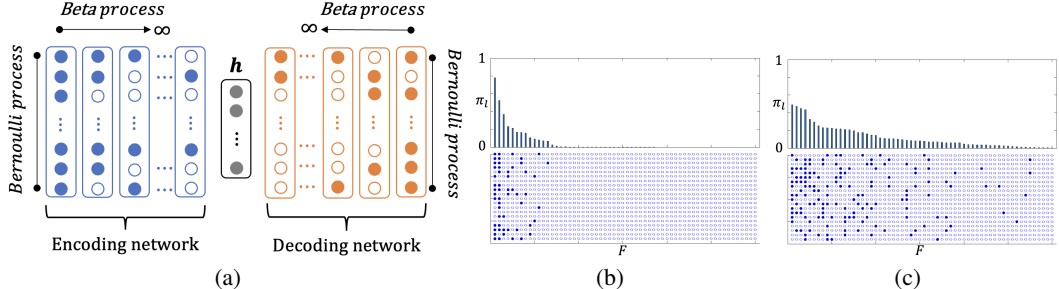

Figure 1: (a): Demonstration of the VAE network structure inference framework. Beta processes induce infinite number of hidden layers for encoding/decodineg networks, and its conjugate Bernoulli process prunes the neurons in each layer with a layer-wise activation probability from the beta process. Filled circles indicate activated neurons, corresponding to a sample of $1$ from the Bernoulli process, while empty circles correspond to deactivated neurons, corresponding to a sample of $0$. (b) and (c): Two settings of the stick-breaking constructions of beta process. The sticks on top are random draws from the process, which act as the layer-wise activation probabilities. A stick location $\delta_{\mathbf{f}_l}$ corresponds to a hidden-layer function $\mathbf{f}_l \in \mathbf{F}$, and the height denotes its activation probability $\pi_l$. The bottom shows the conjugate Bernoulli processes to activate or deactivate neurons in each layer (column-wisely).

incorporates additional constraints to the VAE objective, which enforces a similarity between original input and its semantic preserving transformation in the latent representation [12, 13]. Nonparametric Bayesian inference such as Indian buffet process deep generative models (IBP-DGM) [14] and beta-Bernoulli process VAE (BB-VAE) [15] address overfitting by inferring the dimensionality of VAE latent variables using an Indian buffet process, specifically, a marginalized beta-Bernoulli process prior. While these regularization methods are effective for shallow network structures, they fail to prevent overfitting in deep VAE models.

We propose a novel Bayesian inference framework that automatically adapt VAE network structures by inferring the most plausible encoding/decoding network depths based on the given data, as demonstrated in Figure 1. To achieve this, we employ beta processes [16, 17] to model the number of hidden layers in the encoding/decoding networks, allowing for infinite depths. A conjugate Bernoulli process is utilized to prune the neurons in each layer based on layer-wise activation probabilities generated by the beta process. In addition, to enable efficient joint inference on both the network structures and the latent variables, we extend multiply importance weighted autoencoder (MIWAE) [18] by introducing an additional sample size used for Monte Carlo estimation of the network structures to be tuned. Our theoretical and empirical analysis shows that this novel gradient estimation scheme leads to a tight lower bound with high signal-to-noise ratio of parameter gradients.

In summary, our contributions are: i) We propose AdaVAE, a novel VAE structural adaptation strategy based on Bayesian model selection to enhance model performance. ii) We introduce a scalable estimator that facilitates joint inference on both encoding/decoding network structures and latent variables. iii) We conduct a comprehensive analysis of AdaVAE's regularization capabilities and demonstrate its ability to effectively mitigate overfitting in both shallow and deep VAE models and achieve state-of-the-art performance. iv) We showcase the versatility of AdaVAE by demonstrating its compatibility with different types of VAE backbone networks. It can also be readily applied to various VAE variants, thereby enhancing their performance.

## 2 Related Works

Variational autoencoders (VAEs) have gained popularity as generative models across a wide range of applications [19, 20, 21, 22, 23, 24]. Extensive research efforts have focused on enhancing the performance of VAEs [25, 26, 27, 28, 29, 30, 31, 32]. Notably, well-designed encoding/decoding neural network structures constantly yield state-of-the-art generative performance [4, 5, 6, 33]. Ladder-VAE (LVAE) employs a shared top-down dependency structure in both the inference and generative models to facilitate information sharing between layers [4]. SkipVAE employs skip connections that connect latent variables to each layer in the generative model, mitigating posterior

collapse [34]. BIVA consists of a skip-connected generative model and an inference model formed by a bidirectional stochastic inference path [5]. NVAE designs expressive neural network structures tailored for VAEs based on deep residual networks [6]. Furthermore, advancements in VAE network structure design have also empowered recent works on hybrid models based on VAEs [35, 36, 37, 38].

Current VAE regularization methods often overlook the overfitting effect caused by deep network structures [12, 10, 15, 11, 13]. Amortized inference regularization (AIR) re-interprets the amortized inference model as a regularization for maximum likelihood training. AIR encourages the smoothness of the encoding network to restrict the model capacity, effectively mitigating overfitting and improving predictive performance [10]. Specifically, AIR proposes a denoising variational autoencoder (DVAE) that modifies the VAE objective using random perturbation training. An alternative AIR technique is weight-normalized inference VAE (WNI-VAE) that directly restricts the encoding networks to a set of smooth functions and achieves comparable performance to DVAE. Both approaches exhibit a lower inference gap than standard VAEs [39]. Consistency regularization for VAE (CR-VAE) extends the regularization techniques employed in semi-supervised learning [40, 41] to tackle the inconsistency problem of the inference models by enforcing the latent representations of an image and its semantic-preserving transformation to be similar, thereby yielding robust latent representations accounting for data variation [12].

IBP-DGM [14] and BB-VAE[15] as nonparametric Bayesian inference applied to VAE regularization focus on inferring the dimensionality of latent variable via an Indian Buffet Process (IBP) prior. The IBP prior is derived by marginalization over a beta process, resulting in a binary vector that masks the VAE latent variables. Without accommodating network structures, latent variable regularization alone is inadequate for mitigating overfitting.

## 3 Efficient VAE Estimators

Maximum likelihood estimation of a vector of parameters $\theta$ of a deep generative model $p_\theta(\mathbf{x}, \mathbf{h})$ with $\mathbf{x}$ denoting observed variables and $\mathbf{h}$ denoting latent variables is intractable in general due to the marginalization over $\mathbf{h}$. Amortized variational inference optimizes an evidence lower bound (ELBO) $\mathcal{L}_{\theta,\phi}(\mathbf{x})$ on the log marginal likelihood $\log p_\theta(\mathbf{x}) = \log \int p_\theta(\mathbf{x}, \mathbf{h}) d\mathbf{h}$ by introducing a variational distribution $q_\phi(\mathbf{h}|\mathbf{x})$:

$$
\begin{aligned}
\log p_\theta(\mathbf{x}) &\geq \int q_\phi(\mathbf{h}|\mathbf{x}) \log \frac{p_\theta(\mathbf{x}, \mathbf{h})}{q_\phi(\mathbf{h}|\mathbf{x})} d\mathbf{h} \\
&= \mathbb{E}_{q_\phi(\mathbf{h}|\mathbf{x})}[\log p_\theta(\mathbf{x}|\mathbf{h})] - \mathrm{KL}[q_\phi(\mathbf{h}|\mathbf{x})||p_\theta(\mathbf{h})] \\
&= \mathcal{L}_{\theta,\phi}(\mathbf{x})
\end{aligned}
\tag{1}
$$

where $\phi$ denotes variational parameters. For VAEs, $q_\phi(\mathbf{h}|\mathbf{x})$ denotes an inference model commonly constructed with a neural network. $p_\theta(\mathbf{x}|\mathbf{h})$ denotes a generative model that can also be parameterized by a neural network. The ELBO is optimized with gradient-based methods via reparameterization trick using stochastic Monte Carlo estimators of $\nabla \mathcal{L}_{\theta,\phi}$ [1].

IWAE obtains a tighter lower bound using $K$-sample importance weighting estimate of the log marginal likelihood [42]:

$$
\mathcal{L}_{\theta,\phi}(\mathbf{x}) = \mathbb{E}_{q_\phi(\mathbf{h}|\mathbf{x})} \left[ \log \frac{1}{K} \sum_{k=1}^{K} \frac{p_\theta(\mathbf{x}|\mathbf{h}_k) p_\theta(\mathbf{h}_k)}{q_\phi(\mathbf{h}_k|\mathbf{x})} \right]
\tag{2}
$$

where $\mathbf{h}_k \sim q_\phi(\mathbf{h}|\mathbf{x})$. It shows that the bound gets tighter with increasing $K$. However, [18] presents theoretical and empirical evidence that increasing the importance weighted sample size $K$ to tighten the bound degrades signal-to-noise ratio (SNR) of parameter gradients estimates for the encoding network, and hurts the learning process. A new estimator (MIWAE) is thus introduced to address the issue of diminishing SNR, and its gradient estimate is:

$$
\Delta_{M,K} = \frac{1}{M} \sum_{m=1}^{M} \nabla_{\theta,\phi} \log \frac{1}{K} \sum_{k=1}^{K} \frac{p_\theta(\mathbf{x}|\mathbf{h}_{m,k}) p_\theta(\mathbf{h}_{m,k})}{q_\phi(\mathbf{h}_{m,k}|\mathbf{x})}
\tag{3}
$$

where $\mathbf{h}_{m,k} \sim q_\phi(\mathbf{h}_{m,k}|\mathbf{x})$. For a fixed budget $M \times K$ of total number of hidden variable samples, the number of samples $M$ reduces the variance in estimating the ELBO gradient. $K$ is the importance sample size as in IWAE.

# 4 VAE Structure Inference Framework

Traditional model selection cannot effectively adapt pre-determined VAE network structures to data without incurring significant computation overhead. We thus propose AdaVAE that enables joint inference on the structures of the encoding/decoding networks using stochastic processes [43, 9] and latent variables, and optimizes VAE objective without requiring additional expensive computation.

## 4.1 Formulation of the Inference Model

Let the prior over the latent variables $\mathbf{h}$ be a zero-mean isotropic multivariate Gaussian $p_\theta(\mathbf{h}) = \mathcal{N}(\mathbf{h}; 0, I)$. We formulate the inference model by letting the variational distribution over $\mathbf{h}$ be a multivariate Gaussian with a diagonal covariance structure:

$$q_\phi(\mathbf{h}|\mathbf{x}, \mathbf{Z}) = \mathcal{N}(\mathbf{h}; \boldsymbol{\mu}, \boldsymbol{\sigma}^2 I) \tag{4}$$

where the mean $\boldsymbol{\mu}$ and the standard deviation $\boldsymbol{\sigma}$ are outputs of the encoding neural network $\mathcal{F}_\phi(\mathbf{x})$ with the variational parameters $\phi$. The binary matrix $\mathbf{Z} = [z_{ol} \in \{0, 1\}]$ denotes the network structural variable, as in Figure 1(a).

Let $\mathbf{f}_l$ denote the $l$-th hidden layer of $\mathcal{F}_\phi(\mathbf{x})$ composed of neurons (i.e., non-linear activation functions) $f(\cdot)$. The encoding network $\mathcal{F}_\phi(\mathbf{x})$ has the form:

$$\mathbf{f}_l = f(\mathbf{W}_l \mathbf{f}_{l-1}) \odot \mathbf{z}_{\cdot l} + \mathbf{f}_{l-1} \quad l \in \{1, 2, ..., \infty\} \tag{5}$$

where $\phi = \{\mathbf{W}_l \in \mathrm{R}^{O \times O}\}$, and $\mathbf{W}_l$ is the layer $l$'s weight matrix. $\odot$ denotes element-wise multiplication of two vectors, so that we drop out the $l$-th layer's outputs by multiplying them elementwisely by the column vector $\mathbf{z}_{\cdot l}$ of $\mathbf{Z}$. Each random variable $z_{ol}$ takes the value 1 with $\pi_l \in [0, 1]$ indicating activation probability of the $l$-th layer, as in Figures 1 (b),(c). $O$ is the maximum number of neurons in a layer that is set to be the same for all hidden layers. We have skip connections to propagate the outputs of the hidden layers to the output layer. Note that when this network structure is used as the decoding network for the generative model $p_\theta(\mathbf{x}|\mathbf{h})$, then $\mathbf{h}$ and $\mathbf{x}$ are swapped, and $\theta$ denotes the weights parameters. The output-layer can be readily replaced with a logistic function and the normal distribution with a Bernoulli distribution for binary data.

## 4.2 Beta Process Prior over Layer Number

A beta process $B = \sum_l \pi_l \delta_{\mathbf{f}_l}$, where $\delta_{\mathbf{f}_l}$ is a unit point mass at $\mathbf{f}_l$, is a completely random measure over countably infinite set of pairs $(\mathbf{f}_l, \pi_l)$ [16], where $\mathbf{f}_l \in \mathbf{F}$ denotes a hidden-layer function and $\pi_l$ is its activation probability $\pi_l \in [0, 1]$. Its conjugate Bernoulli process can be defined as $\mathbf{Z}_{o\cdot} \sim \mathrm{BeP}(B)$, where $\mathbf{Z}_{o\cdot} = \sum_l z_{ol} \delta_{\mathbf{f}_l}$ is at the same locations $\delta_{\mathbf{f}_l}$ as $B$ where $z_{ol}$ are independent Bernoulli variables with $\pi_l$ being the probability of $z_{ol} = 1$. As in Eqn. (5), $z_{ol} = 1$ activates the $o$'th neuron in layer $l$. Computationally, we employ the stick-breaking construction [17] of beta process and its conjugate Bernoulli process as

$$z_{ol} \sim \mathrm{Ber}(\pi_l), \quad \pi_l = \prod_{j=1}^{l} \nu_j, \quad \nu_l \sim \mathrm{Beta}(\alpha, \beta) \tag{6}$$

where $\nu_l$ are sequentially drawn from a beta distribution. The hyperparameters $\alpha$ and $\beta$ can be set to balance the network depth and width. Specifically, Figure 1 (b) demonstrates that if $\beta > \alpha > 1$, the network structure prior favors shallower but wider network structures with the first few layer-wise activation probabilities being high. If $\alpha > \beta > 1$, activation probabilities tend to be low over larger number of active layers, and the prior prefers a deeper but narrower network, as in Figure 1 (c).

We thus define the prior over the encoding network structural variable $\mathbf{Z}$ as

$$p_{\alpha,\beta}(\mathbf{Z}, \boldsymbol{\nu}) = p_{\alpha,\beta}(\boldsymbol{\nu})p(\mathbf{Z}|\boldsymbol{\nu}) = \prod_{l=1}^{\infty} \mathrm{Beta}(\nu_l|\alpha, \beta) \prod_{o=1}^{O} \mathrm{Ber}(z_{ol}|\pi_l) \tag{7}$$

To enable asymmetric encoding/decoding network structures, we independently apply the prior to both networks. Further analysis on symmetric constraints can be found in the Appendix.

## 4.3 Joint Inference on VAE Network Structures and Latent Variables

We first expand the overall marginal likelihood over the VAE structures $\mathbf{Z}$ as

$$\log p_{\alpha,\beta}(\mathbf{x}) = KL[q(\mathbf{Z},\boldsymbol{\nu}|\{a_t\}_{t=1}^T,\{b_t\}_{t=1}^T)||p_{\alpha,\beta}(\mathbf{Z},\boldsymbol{\nu}|\mathbf{x})] + \mathcal{L}_{\{a_t\},\{b_t\}}(\mathbf{x}) \qquad (8)$$

where the first RHS term denotes a Kullback-Leibler (KL) divergence of the approximate variational distribution from the true posterior of the VAE network structural variables. We specify the variational distribution as

$$q(\mathbf{Z},\boldsymbol{\nu}|\{a_t\}_{t=1}^T,\{b_t\}_{t=1}^T) = \prod_{t=1}^T \text{Beta}(\nu_t|a_t,b_t) \prod_{o=1}^O \text{ConBer}(z_{ot}|\pi_t) \qquad (9)$$

where $\pi_t = \prod_{j=1}^t \nu_j$, and $\{a_t, b_t\}_{t=1}^T$ are the variational parameters. $T$ denotes a truncation level for the maximum number of hidden layers [17]. We also relax the constraint of the discrete variables by reparameterizing the Bernoulli distribution into a concrete Bernoulli distribution $\text{ConBer}(z_{ot}|\pi_t)$ [44, 45]. This allows us to efficiently backpropagate the parameter gradients of the estimator while generating network structure samples.

The second RHS term in Eqn. (8) denotes the ELBO to the overall marginal likelihood:

$$\mathcal{L}_{\{a_t\},\{b_t\}}(\mathbf{x}) = \int q(\mathbf{Z},\boldsymbol{\nu}|\{a_t\}_{t=1}^T\{b_t\}_{t=1}^T)(\log p_\theta(\mathbf{x}|\mathbf{Z}) + \log p_{\alpha,\beta}(\mathbf{Z},\boldsymbol{\nu})$$
$$- \log q(\mathbf{Z},\boldsymbol{\nu}|\{a_t\}_{t=1}^T\{b_t\}_{t=1}^T))d\mathbf{Z}d\boldsymbol{\nu} \qquad (10)$$

The term $\log p_\theta(\mathbf{x}|\mathbf{Z})$ in Eqn. (10) is the marginal likelihood over the latent variable $\mathbf{h}$, which is an extension of Eqn. (1), in terms of conditioning on the structure variable $\mathbf{Z}$. Thus, the ELBO to the marginal likelihood $\log p_\theta(\mathbf{x}|\mathbf{Z})$ is:

$$\log p_\theta(\mathbf{x}|\mathbf{Z}) \geq \int q_\phi(\mathbf{h}|\mathbf{x},\mathbf{Z}) \log \frac{p_\theta(\mathbf{x},\mathbf{h}|\mathbf{Z})}{q_\phi(\mathbf{h}|\mathbf{x},\mathbf{Z})}d\mathbf{h}$$
$$= \mathbb{E}_{q_\phi(\mathbf{h}|\mathbf{x},\mathbf{Z})}[\log p_\theta(\mathbf{x}|\mathbf{h},\mathbf{Z})] - KL[q_\phi(\mathbf{h}|\mathbf{x},\mathbf{Z})||p_\theta(\mathbf{h}|\mathbf{Z})]$$
$$= \mathcal{L}_{\theta,\phi}(\mathbf{x}|\mathbf{Z}) \qquad (11)$$

**Lemma 1** *Let $Q(\mathbf{h}|\mathbf{x}) = \int q_\phi(\mathbf{h}|\mathbf{x},\mathbf{Z})q(\mathbf{Z},\boldsymbol{\nu})d\mathbf{Z}d\boldsymbol{\nu}$ be the variational distribution of the latent variable $\mathbf{h}$ marginalizing over $\mathbf{Z}$, then*

$$\mathbb{E}_{q(\mathbf{Z},\boldsymbol{\nu})}\mathbb{E}_{q_\phi(\mathbf{h}|\mathbf{x},\mathbf{Z})} \log \frac{p_\theta(\mathbf{x},\mathbf{h}|\mathbf{Z})}{q_\phi(\mathbf{h}|\mathbf{x},\mathbf{Z})} \leq \mathbb{E}_{Q(\mathbf{h}|\mathbf{x})} \log \frac{p_\theta(\mathbf{x},\mathbf{h}|\mathbf{Z})}{Q(\mathbf{h}|\mathbf{x})} \qquad (12)$$

Lemma 1 indicates that the overall ELBO we derived on the left-hand side in Eqn. (12) bounds the lower bound to the marginal likelihood over $\mathbf{h}$. The proof is in the appendix. In particular, $q(\mathbf{Z},\boldsymbol{\nu})$ is essentially a non-explicit mixing distribution [46]. It allows the variational distribution $Q(\mathbf{h}|\mathbf{x})$ to take complex form, and results in more informative latent representation.

We adopt Monte Carlo estimation of the expectations over both $\mathbf{Z}$ in Eqn. (10) and $\mathbf{h}$ in Eqn. (11) to estimate the overall ELBO. In particular, we extend the MIWAE estimator in Eqn. (3), and introduce three sample sizes to tune: the number of samples $S$ used for Monte Carlo estimation of the expectation over the VAE network structure variable $\mathbf{Z}$, the number of samples $M_s$ used for Monte Carlo estimation of the gradient of the latent variable ELBO conditioned on the structure samples in Eqn. (11), and the number of importance samples $K_s$ used for estimation of the expectation over the latent variables $\mathbf{h}$. We thus express our gradient estimate in the general form as

$$\Delta_{S,M,K} = \frac{1}{S}\sum_{s=1}^S \frac{1}{M_s}\sum_{m=1}^{M_s} \nabla_{\theta,\phi} \log \frac{1}{K_s}\sum_{k=1}^{K_s} \frac{p_\theta(\mathbf{x}|\mathbf{h}_{m,k},\mathbf{Z}_s)p_\theta(\mathbf{h}_{m,k}|\mathbf{Z}_s)}{q_\phi(\mathbf{h}_{m,k}|\mathbf{x},\mathbf{Z}_s)} \qquad (13)$$

where $\mathbf{h}_{m,k} \sim q_\phi(\mathbf{h}|\mathbf{x},\mathbf{Z})$. When $S = 1$ our estimator becomes equivalent to the MIWAE objective in Eqn. (3). Since our estimator generates latent variable samples conditioned on network structure samples, increasing $S$ will not impact the SNR of the gradient estimator in Eqn. (13).

**Theorem 1** *Let $\mathcal{L}_S$ be the lower bound with $S$ structure samples of $\mathbf{Z}_s \sim q(\mathbf{Z},\boldsymbol{\nu})$, then:*

$$\mathcal{L}_S \leq \mathcal{L}_{S+1} \leq \log p_{\alpha,\beta}(\mathbf{x}), \mathcal{L}_S = \mathbb{E}_{q(\mathbf{Z},\boldsymbol{\nu})}\mathbb{E}_{q_\phi(\mathbf{h}|\mathbf{x},\mathbf{Z})} \log[\frac{1}{S}\sum_s \frac{p_\theta(\mathbf{x},\mathbf{h}|\mathbf{Z_s})}{q_\phi(\mathbf{h}|\mathbf{x},\mathbf{Z_s})}] \qquad (14)$$

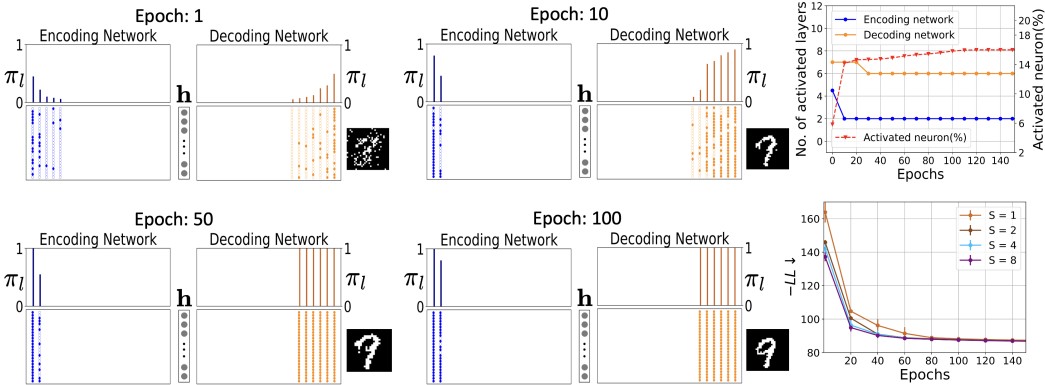

Figure 2: Left: Evolution of the encoding/decoding network structures visualized through layer-wise activation probabilities $\pi_l$ (top) and neuron activations $\mathbf{Z}$ (bottom) with a reconstructed sample. Right: the top shows the median change of the number of active layers over training epochs and the percentage of activated neurons in the truncation. The bottom shows the convergence of the proposed estimator in terms of negative log-likelihood ($-LL$) for different structure sample size $S$.

Table 1: Test performance in negative log-likelihood ($-LL$) mean $\pm 1$ standard deviation (lower the better) over $4$ runs with random initialization. The overall best result on each dataset is bolded.

| Dataset | (M,K) | S=1 | S=2 | S=4 | S=8 |
|---------|-------|-----|-----|-----|-----|
| MNIST | (8,8) | **82.25**$_{\pm\mathbf{0.05}}$ | 82.50$_{\pm 0.00}$ | 82.60$_{\pm 0.10}$ | 82.30$_{\pm 0.10}$ |
|  | (4,16) | 82.47$_{\pm 0.45}$ | 82.33$_{\pm 0.13}$ | 82.52$_{\pm 0.02}$ | 83.02$_{\pm 0.20}$ |
| Omniglot | (8,8) | 107.10$_{\pm 0.10}$ | 106.45$_{\pm 0.10}$ | 106.55$_{\pm 0.30}$ | **106.34**$_{\pm\mathbf{0.01}}$ |
|  | (4,16) | 108.12$_{\pm 0.16}$ | 107.15$_{\pm 0.08}$ | 107.35$_{\pm 0.40}$ | 108.30$_{\pm 0.50}$ |
| Caltech101 | (8,8) | 116.83$_{\pm 1.57}$ | 114.94$_{\pm 0.45}$ | 114.00$_{\pm 0.42}$ | 113.54$_{\pm 0.40}$ |
|  | (4,16) | 116.30$_{\pm 1.11}$ | 114.55$_{\pm 1.18}$ | 113.02$_{\pm 0.34}$ | **112.53**$_{\pm\mathbf{0.42}}$ |

Proof of this theorem is in the appendix. The theorem shows the convergence of our estimator. Specifically, increasing $S$ leads to a tighter lower bound for the overall marginal likelihood. Training an encoding/decoding network with depth $L$ and width $M$, the time complexity is $T_c = O(NBLM^2)$ with $N$ training examples and $B$ epochs. Our method is linearly scalable as $ST_c$. With a proper thresholding, the number of active layers $L$ is relatively small in each sample.

## 5   Experiments

We analyze the behavior of our inference framework across various tasks. We study how AdaVAE facilitates the evolution of encoding/decoding network structures for inferring the most plausible depth from the given data, while generating expressive latent representations. Next, we explore the impact of the structure sample size $S$ on the convergence of the proposed estimator in Eqn.(13). Then we show that AdaVAE effectively mitigates overfitting in both shallow and deep network settings, leading to state-of-the-art performance on benchmark datasets. Finally, we demonstrate the framework's compatibility with different types of backbone networks and VAE variants. [2]

### 5.1   Adaptive VAE Network Structures

AdaVAE enables us to perform joint inference on both encoding/decoding network structures and latent variables. To investigate how network structures evolve during training epochs, we set the truncation level $T = 25$ on MLP backbone nets with $tanh$ non-linearities. We analyze adaVAE's behavior on $28 \times 28$ binarized MNIST images [47], employing structure sample sizes $S = \{1, 2, 4, 8\}$. We run the experiments 3 times and averaging the outcomes. Figure 2 Left shows the evolution of the encoding/decoding network structures for one trial with $S = 8$. AdaVAE initializes multiple hidden

---

[2]Implementation details are in the Appendix. Codes are provided.

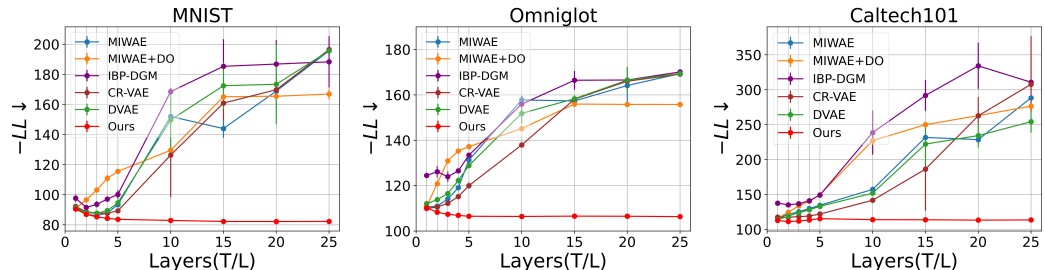

Figure 3: The performance of VAE regularization methods changes with network depths. Our proposed method effectively prevents overfitting for both small and large truncations $T$, consistently achieving the best performance. In contrast, for $1 \leq L \leq 5$, the performance of other regularization methods initially improves but then starts to decline, suggesting they suffer from overfitting even for shallow structures.

Table 2: The best performance of our method and the VAE regularization methods in Figure 3. We also demonstrate the compatibility of our framework.

| | MNIST | | Omniglot | | Caltech 101 | |
|---|---|---|---|---|---|---|
| | -LL $\downarrow$ | MI $\uparrow$ | -LL $\downarrow$ | MI $\uparrow$ | -LL $\downarrow$ | MI $\uparrow$ |
| MIWAE [18] | $86.11_{\pm0.01}$ | $\mathbf{9.13}_{\pm\mathbf{0.01}}$ | $110.61_{\pm0.10}$ | $\mathbf{8.98}_{\pm\mathbf{0.00}}$ | $116.19_{\pm0.08}$ | $7.64_{\pm0.01}$ |
| MIWAE+DO | $90.99_{\pm0.01}$ | $\mathbf{9.13}_{\pm\mathbf{0.01}}$ | $110.89_{\pm0.01}$ | $\mathbf{8.98}_{\pm\mathbf{0.00}}$ | $116.00_{\pm0.40}$ | $7.63_{\pm0.01}$ |
| DVAE [10] | $87.67_{\pm0.17}$ | $9.07_{\pm0.00}$ | $112.04_{\pm0.13}$ | $8.96_{\pm0.00}$ | $113.71_{\pm0.43}$ | $7.53_{\pm0.01}$ |
| CR-VAE [12] | $87.67_{\pm0.05}$ | $9.03_{\pm0.01}$ | $109.94_{\pm0.19}$ | $8.97_{\pm0.00}$ | $117.46_{\pm0.10}$ | $7.37_{\pm0.01}$ |
| IBP-DGM [14] | $92.24_{\pm0.75}$ | - | $124.01_{\pm1.62}$ | - | $135.23_{\pm0.54}$ | - |
| BB-VAE [15] | $91.55_{\pm0.69}$ | - | $124.47_{\pm0.58}$ | - | $135.18_{\pm0.83}$ | - |
| Ours | $\mathbf{82.30}_{\pm\mathbf{0.10}}$ | $\mathbf{9.13}_{\pm\mathbf{0.01}}$ | $106.34_{\pm0.01}$ | $\mathbf{8.98}_{\pm\mathbf{0.00}}$ | $113.54_{\pm0.40}$ | $\mathbf{7.67}_{\pm\mathbf{0.02}}$ |
| Ours+DVAE | $83.30_{\pm0.20}$ | $9.07_{\pm0.02}$ | $107.80_{\pm0.40}$ | $8.96_{\pm0.00}$ | $111.92_{\pm0.37}$ | $7.63_{\pm0.00}$ |
| Ours+CR-VAE | $85.20_{\pm0.20}$ | $9.03_{\pm0.01}$ | $\mathbf{105.60}_{\pm\mathbf{0.05}}$ | $8.97_{\pm0.00}$ | $\mathbf{108.93}_{\pm\mathbf{1.40}}$ | $7.51_{\pm0.02}$ |

layers with sparsely activated neurons, gradually converging to fewer fully activated layers. Figure 2 right top presents the medians of the number of active hidden layers in the encoding/decoding networks, as well as the percentage of activated neurons in the truncation changing over epochs. The encoding network structure stabilizes at two active layers, while the decoding network settles at six active layers. The decoding network tends to have more active layers compared to the encoding network on Omniglot as well (see Appendix). Figure 2 right bottom shows the generative performance assessed by negative log-likelihood ($-LL$) converge faster with an increased structure sample size $S$, which is consistent with **Theorem 1**.

## 5.2 Effect of Structure Sample Sizes

We assess our estimator using three benchmark datasets: MNIST [47], Omniglot [48], and Caltech101 Silhouettes [49]. In each minibatch, we set a budget of $S \times M_s \times K_s = 64$ total latent variable samples for each datapoint. We examine four settings of the VAE structure sample size $S = \{1, 2, 4, 8\}$, along with latent variable sample sizes $(M, K) = \{(8, 8), (4, 16)\}$ as in [18]. The truncation level is $T = 25$ with a maximum width $O = 200$. The distribution over the output from the decoding networks is factorized Bernoulli. Table 1 indicates larger values of $S$ generally yield better performance. For MNIST, there is no statistically significant difference between $S = 1$ and $S = 8$. Among the two structure sample sizes, the best importance sample configuration is $(M, K) = (8, 8)$.

## 5.3 On Preventing Overfitting

To assess the performance of our method across varying truncation level $T$, we compare with existing VAE regularization methods: Denoising VAE (DVAE) [10], CR-VAE [12], and BB-VAE [15], MIWAE with dropout (MIWAE+DO), along with vanilla MIWAE [18]. as in Figure 3. All methods share the same maximum width of $O = 200$ and a latent variable dimensionality of 50. For

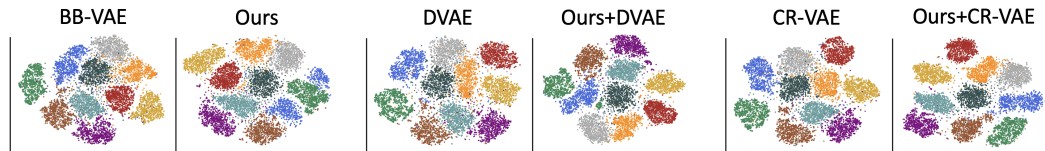

Figure 4: Visualization of the latent representation via t-SNE embeddings for the MNIST dataset. The embeddings are colored based on class labels. DVAE and CR-VAE combined with our framework result in better representations.

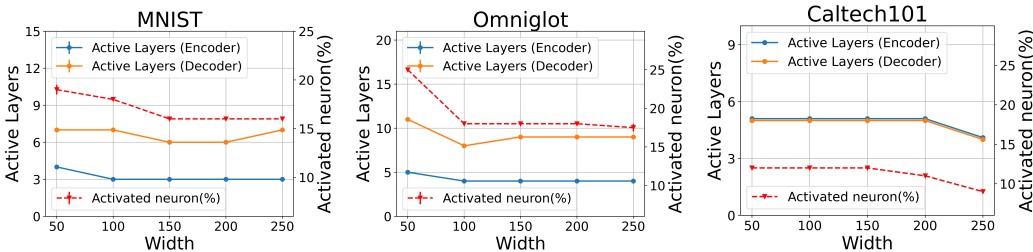

Figure 5: Influence of the maximum number of neurons per layer $O$ on our method. When $O$ is small (e.g., $O \leq 100$), we tend to have shallower encoding/decoding networks. As $O$ becomes reasonably large (e.g., $O \geq 100$), it tends not to have significant influence on the depth. Meanwhile, the percentages of activated neurons in the truncation become stable.

IBP-DGM and BB-VAE, we set the maximum dimensionality of the latent variables to $100$, following the setup of [15]. In Figure 3, we set depth $L$ for the regularization methods and truncation level $T$ for our method over the range $L = T = \{1, 2, 3, 4, 5, 10, 15, 20, 25\}$, allowing us to compare their effectiveness in both shallow (i.e., $L = T \leq 5$) and deep (i.e., $L = T \geq 5$) VAE network structures.

Figure 3 shows that when $L = T \leq 5$, the VAE regularization methods can mitigate overfitting. However, even for the shallow cases as $L = T = \{4, 5\}$, the performance of these methods is affected by overfitting, as evidenced by the "U"-shaped performance curves, indicating the classical variance-bias trade-off. Despite our method exhibiting slight underfitting issues for $L = T \leq 2$, it still outperforms other methods. For deep VAE structures with $T/L = \{10, 15, 20, 25\}$, Figure 3 shows that our method's performance is minimally impacted by large truncations. This robustness indicates our method's ability to mitigate overfitting for large VAE structure settings. In contrast, as the depth $L$ increases, the steadily increasing negative log-likelihood ($-LL$) for other regularization methods suggests their inability to prevent overfitting in deep VAE models. Overall, AdaVAE consistently delivers superior performance for both shallow and deep VAE structures. An analysis of the computational time required by our framework and its comparison to the baselines is presented in the Appendix.

Table 2 highlights our superiority over other methods in terms of density estimation and mutual information (MI) [50]. Additionally, when incorporating our framework with DVAE through the addition of Gaussian noise to the input or with CR-VAE by integrating the consistency regularization term into our estimator, we observe enhanced performance on Omniglot and Caltech101 datasets. Furthermore, in Figure 4, we visualize the latent representations of the VAE methods. DVAE and CR-VAE when combined with our framework result in well-separated clusters, indicating that application of our framework allows VAEs to learn a meaningful latent representation. A quantitative evaluation of the latent representations is presented in the Appendix by analyzing the performance of VAE methods on a downstream classification task.

## 5.4 Effects of the Maximum Width $O$

We further investigate the influence of the width $O$ (i.e., the maximum number of neurons per layer). Figure 5 shows the evolution of the medians of the number of active hidden layers (i.e., the hidden layers with activated neurons) as $O$ increases. When $O \leq 100$, we tend to have shallower

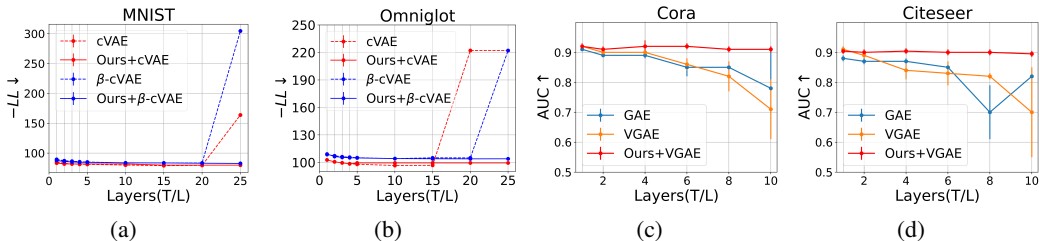

Figure 6: Our framework's performance on different VAE backbone networks. (a) and (b) show VAE and $\beta$-VAE [51] with convolutional layers on MNIST and Omniglot datasets. (c) and (d) show VGAEs [52] with graph convolutional layers on Cora and Citeseer datasets.

encoding/decoding networks with a lower percentage of activated neurons in the truncation. However, when $O$ is reasonably large as $O \geq 100$, it has no influence on the network depth. In particular, the percentage of activated neurons in the whole truncation also remains relatively stable. This suggests that our method can automatically balance network depth and width to maintain the best performance.

## 5.5 Application to VAE Backbone Networks

We demonstrate adaVAE's efficacy by applying it to VAEs with different types of encoding/decoding backbone networks. To infer the number of convolutional layers in a convolutional VAE (cVAE) using the beta process, we readily mask the convolution channels in layer $l$ with $\mathbf{z}_{\cdot l}$. Figure 6 (a) and (b) show that by adapting network structures to the data we improve the overall performance of cVAE and $\beta$-cVAE [51]. The adaptive backbone networks effectively prevent overfitting for deep structure settings (i.e., $T/L \geq 15$).

Variational graph autoencoder (VGAE) [52] encodes graph nodes to latent embeddings with graph convolutional (GC) layers and re-creates the input graph from the embeddings by predicting the existence of an edge between the nodes. We combine our framework with VGAE by elementwisely multiplying the GC channels (i.e., the feature vectors) of layer $l$ with $\mathbf{z}_{\cdot l}$ in both its encoding/decoding networks. We compare the hybrid method's performance with graph autoencoders (GAEs) [52] and vanilla VGAEs on two benchmark graph-structured datasets: Cora [53] and Citeseer [54]. The AUC scores of link prediction over varying numbers of GC layer settings are shown in Figure 6 (c) and (d). Our framework enables the VGAE to maintain its best performance for all the network depth settings by automatically adapting its structures to the data, whereas the performance of GAE and vanilla VGAE drops with the increase of the layer numbers (e.g., $T/L = \{6, 8, 10\}$).

## 5.6 Application to VAE Variants

We assess the performance of our inference framework by leveraging it to adapt the network structures of VAE variants to data. Specifically, for $\beta$-VAE, we apply layer-wise binary masks $\mathbf{z}_{\cdot l}$ to the convolutional channels and infer the layer numbers using the beta process. In the case of Ladder-VAE (LVAE) [4], we adjust its depth by applying layer-wise binary masks to its deterministic layers in the bottom-up dependency structures and add skip connections between the stochastic layers. For SkipVAE [34], we model its depth by employing layer-wise binary masks and skip connections in both its encoding/decoding networks. The expressive network structures of NVAE [6] consists of multiple blocks of convolutional layers. We apply our

Table 3: Performance comparison of VAE variants with and without our inference framework in terms of $-LL$, MI, and KL divergence.

| Methods | -LL $\downarrow$ | MI $\uparrow$ | KL $\uparrow$ |
|---|---|---|---|
| $\beta$-cVAE ($\beta = 2$) [51] | $106.50_{\pm 0.12}$ | $8.46_{\pm 0.01}$ | $18.01_{\pm 0.21}$ |
| Ours + $\beta$-cVAE ($\beta = 2$) | $\mathbf{102.45}_{\pm 0.07}$ | $\mathbf{8.47}_{\pm 0.01}$ | $\mathbf{20.65}_{\pm 0.02}$ |
| LVAE [4] | $136.50_{\pm 1.50}$ | $8.43_{\pm 0.02}$ | $20.05_{\pm 0.21}$ |
| Ours+LVAE | $\mathbf{121.48}_{\pm 0.67}$ | $\mathbf{8.50}_{\pm 0.01}$ | $\mathbf{21.00}_{\pm 0.47}$ |
| SkipVAE [34] | $112.68_{\pm 0.80}$ | $8.50_{\pm 0.01}$ | $22.82_{\pm 0.79}$ |
| Ours+SkipVAE | $\mathbf{108.00}_{\pm 0.04}$ | $8.50_{\pm 0.01}$ | $\mathbf{28.26}_{\pm 0.10}$ |
| NVAE [6] | $\mathbf{98.83}_{\pm 0.17}$ | $8.51_{\pm 0.01}$ | $34.97_{\pm 0.13}$ |
| Ours+NVAE | $99.10_{\pm 0.20}$ | $8.51_{\pm 0.01}$ | $\mathbf{37.85}_{\pm 0.50}$ |

framework to infer the number of blocks for a light NVAE version without top-down dependency between the blocks. The detailed settings of these VAE variants and additional results can be found in Appendix. The results in Table 3 on FashionMNIST [55] demonstrate that by inferring the encoding/decoding network structures we significantly improve the density estimation performance of the VAE variants. Our framework also boosts their ability to mitigate posterior collapse as indicated by MI and KL divergence.

## 6 Conclusion

We present a Bayesian inference framework and a scalable estimator that automatically adapts VAE network structures to data. The experiments demonstrate its effectiveness in preventing overfitting for both shallow and deep structure settings. Moreover, AdaVAE exhibits promising applications across various types of VAE backbone networks and VAE variants, including those with hierarchical structures such as LVAE. Notably, AdaVAE enhances the generative performance of these models without requiring pre-determined network structures prior to training. Our future work entails relaxing the constraint of truncation levels by incorporating the Russian roulette method [56] and scaling up the inference for large images.

## 7 Acknowledgements

This material is based upon work supported by the National Science Foundation under NSF Award No.2045804 and Award No.1850492. We are thankful to Kishan KC for helpful discussion. We acknowledge Research Computing at the Rochester Institute of Technology [57] for providing computational resources.

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

# A  Appendix

This section is in the supplemental material.

