# OpenReview forum: "AdaVAE: Bayesian Structural Adaptation for Variational Autoencoders"
_NeurIPS.cc/2023/Conference — NeurIPS 2023 poster_

### Official Review · Reviewer_kqwq · 2023-07-04

**Soundness:** 3 good
**Presentation:** 3 good
**Contribution:** 2 fair
**Rating:** 6
**Confidence:** 5

**Summary:**

The paper proposes a structure adaptation algorithm specifically tailored to the well-known Variational Autoencoders (VAEs). The main motivation lies in the fact that the structure of a generative model plays a significant part in the overall performance, something that is very under-explored in related VAE literature. To this end, the authors turn to the solid Bayesian inference framework and utilize a structural adaptation approach based on the Beta Process to infer the optimal network depth and the Bernoulli process to prune neurons in each hidden layer. The experimental evaluation focuses on the adaptation abilities of the proposed framework, how the structure sample size affects convergence and its over-fitting prevention properties. Finally the authors discuss how to integrate the proposed approach on VAE backbones and other VAE variants in general.

**Strengths:**

This work focuses on a very interesting aspect of modern architectures, that is their structure. The paper is overall easy to follow. The notation is clear and almost everything is well defined.
By now there has been a lot of work in the community that aim to address this challenge through different views such as pruning approaches, component omission mechanisms, e.t.c.  Compared to most of existing approaches the above formulation allows for simultaneously inferring both the network depth and width of each layer in a principled way via Bayesian arguments.

**Weaknesses:**

1) As the authors note, the idea of using a Beta-Bernoulli pair in order to adapt the model capacity is not something new. There exist several works in both discriminative and generative models that aim to tackle this issue [1,2,3,4,5]. The authors however fail to appropriately introduce and discuss the differences with and advantages/drawbacks compared to these different methods leading to a single two sentence mention of one of the most similar approaches, i.e., BB-VAE. Instead the authors seem to focus more to the regularization aspect of other dissimilar methods.

2) In this context, the work presented in [4] should be the main focus on comparison because it is essentially the same exact method, with minor adaptations for the VAE structure. Apart from the difference in estimation, i.e., Gumbel Softmax vs MIWAE, the differences are minor and down to some notation changes. The authors do cite the paper at some point  without any kind of discussion.

3) Taking into consideration that the method is very similar to [4], a core difference is the different estimator. Did the authors investigate the impact of other estimation methods? How does it compare to Gubmel Softmax in terms of both complexity and convergence?

4) The authors should briefly expand on the chosen metrics in the respective tables. The negative log likelihood is a well known measure, while the MI and KL divergence can have substantially different interpretations. The authors cite the work of [6], which dissects the ELBO into three different terms that probably correspond to the Tables 2,3. Further clarifications for the reader are essential. For example, why is the largest KL the better? The purpose of optimization is the minimization of the KL divergence and in [6] it is also noted that "whenever it is large it indicates a very strong and potentially unwanted regularization effect from the prior"

5) The stability of the training process might be an issue that isn't addressed in the main text, especially in the stick-breaking construction. Did issues arise during training due to very small or very large values? and how did the authors address them?

6) There is no discussion about the computational and memory complexity of the approach. It is apparent that the introduction of the additional latent parameters and the KL terms will significantly contribute to the overall footprint of the method, especially with multiple samples due to the MIWAE estimation. Some wall time measurements are necessary for both training and inference.

7) There is not a clear definition of the prediction process of the framework. Do you draw multiple samples from the learned posteriors and sample the active ones? do the authors use a threshold in "each sample"? (line 182)

8) The performance of the method on different VAR backbones is not clear. The authors mention the masking of convolutional channels in order to infer the number of convolutional layers, without any further expansion on the specific formulation and results. Does adaptation take place in specific features maps similar to [2] or the options are to drop the layer or not?

9) Apart from the t-SNE visualization, some reconstruction comparisons could be useful.

Considering these points and the similarity of this work to [4], I believe that the novelty and overall contribution of this work is very limited for publication.

[1] Sotirios P Chatzis. Indian buffet process deep generative models for semi-supervised classification. In 2018
IEEE International Conference on Acoustics, Speech and Signal Processing (ICASSP)

[2] Konstantinos Panousis, Sotirios Chatzis, and Sergios Theodoridis. Nonparametric bayesian deep networks with
local competition. In International Conference on Machine Learning, pages 4980–4988. PMLR, 2019.

[3] Rachit Singh, Jeffrey Ling, and Finale Doshi-Velez. Structured variational autoencoders for the beta-
bernoulli process. In NIPS 2017 Workshop on Advances in Approximate Bayesian Inference, 2017.

[4] Kishan KC, Rui Li, and Mohammad Mahdi Gilany. Joint inference for neural network depth and dropout
regularization. In Advances in Neural Information Processing Systems (NeurIPS), volume 34, 2021.

[5] Xu, W., Chen, R., Li, X., & Duvenaud, D. (2022). Infinitely Deep Bayesian Neural Networks with Stochastic Differential Equations. International Conference on Artificial Intelligence and Statistics.

[6] Matthew D Hoffman and Matthew J Johnson. ELBO surgery: Yet another way to carve up the variational evidence lower bound. In Workshop in Advances in Approximate Bayesian Inference, NeurIPS, volume 1, 2016.

**Questions:**

Please see the Weaknesses section.

**Limitations:**

Please see Weaknesses, especially concerning complexity and stability.

---

> ### Author Rebuttal · Authors · 2023-08-09
>
> **Novelty of our framework comparing with other beta-Bernoulli based methods**: Our work diverges fundamentally from [1,2,3] by treating the expansion of VAE network structures as a stochastic process within a comprehensive framework, in contrast to their exclusive focus on the regularization of VAE latent variable, overlooking network structures. Moreover, our experiments in Figure 3 distinctly demonstrate that adapting network architectures is a notably more effective strategy to boost VAEs' performance than the one-size-fits-all methods. In comparison to [4], which infers feedforward network depths in supervised learning scenarios, we have innovatively introduced a novel generative learning approach tailored to diverse VAE variants. The novelty contribution of our VAE structural adaptation framework is non-trivial, because 1) we developed a new estimator to jointly infer both net structures and latent variables. 2) We also enabled the asymmetric encoding/decoding network structures in the inference procedure. 3) Our extensive experiments show our framework and our estimator boosted different VAE backbone networks' and various VAE variants' performance and achieved state-of-the-art.
>
> **Gumbel-softmax vs MIWAE**: Gumbel-softmax is a reparameterization trick, relaxing categorical distributions to continuous ones. It is not a VAE estimator we can employ or compare with. The technique is not involved or relevant to our work.
>
> **Discussion of chosen metrics**: We have a detailed discussion on the evaluation metrics in Appendix Section 5.1.
>
> **Stability with training**: We do not have any stability issues with our training algorithm. We provided a theoretical analysis of the algorithm in Theorem 1 and its proof in Appendix Section 2. We analyzed its convergence in training in Appendix Section 6. It shows no stability issues. We have the pseudocode in the Appendix Section 4 and we've also included the codes in the Supplementary Material.
>
> **Computational and memory complexity**: We have computational complexity analysis in Line 179-182. We also provided detailed comparison of the running times of the VAE variant with/without our framework in the Supplementary Material/Appendix Section 8.
>
> **Prediction process**: We described the prediction process in the Supplementary Material/Appendix Line 23-24. For prediction, we compute the ELBO i.e the IWAE  estimator in Eqn. (2) using 5000 importance samples and 1 structure sample. We will move it to the camera-ready version of our paper.
>
> **Performance on VAE variants**: We detailed the implementation of combining our framework with VAE variants in Appendix Section 5.5.
>
> **Reconstruction evaluation**: We demonstrated reconstructions in Appendix Section 10.1 (Figures 7 and 8) and provided qualitative comparison and analysis.

---

> > ### Author Response · Authors · 2023-08-14
> > **Additional points on the rebuttal**
> >
> > We added a couple of points to address Reviewer kqwq's concerns.
> >
> > **Comparison with [2]**: The work presented in [2] is primarily centered around reducing network complexity by inferring the Local Winner Takes All (LWTA) connections with Indian Buffet Process (IBP) in order to regularize the network size. It's important to note that this study still needs to pre-specify a fixed depth for the neural network prior to training.  Also, the method solely demonstrates the proposed method in supervised learning setting. Notably, IBP can be derived from a marginalized version of beta-Bernoulli process. However, the key difference of our approach of utilizing beta-Bernoulli processes lies in the separate application of them. More precisely, we employ the beta process to model the growth of the encoding/decoding network depth, and regularize their width with the Bernoulli process, respectively. We will include the reference and our discussion in the revised version.
> >
> > **Comparison with [5]**: The infinite parameters in [5] is introduced by continuous neural networks based on differential equations. The application of their approach is confined to supervised learning scenarios, and it remains unclear how this technique can be extended to encompass VAE and their variants within unsupervised learning context. The method also has some limitations comparing with our work e.g., it requires Lipschitz property to guarantee a unique solution. We concur with the reviewer it is a good idea to further investigate how this class of methods could be applied on VAEs with different backbone networks. It will become part of our future research. We will cite and discuss this reference in the revised version of our manuscript.

---

> > > ### Comment · Reviewer_kqwq · 2023-08-14
> > >
> > > I thank the authors for (both) their responses to my concerns and to the concerns of the other reviewers.
> > >
> > > It is important to highlight that I acknowledge the contributions of the authors and I do appreciate the consideration of a principled Bayesian inference framework, instead of relying on ad-hoc metrics and heuristics to adapt the network structure.
> > >
> > > However, I found that the original submission was missing in terms of presentation and comparison with related non-parametric Bayesian work and instead focused on comparisons with "less" related regularization methods. Even though other works consider the IBP and other frameworks as regularization or complexity reduction measures, they are in principle adapting the structure (usually in terms of width only). Thus, discussing and comparing these works as the authors did in the rebuttal phase is very critical for the overall presentation. I find the discussion with the authors very fruitful on this aspect and I suggest that the authors restructure the related work to reflect this.
> > >
> > > Moreover, as emerged from the authors' rebuttal, key details of the approach are found in the Supplementary Material such as the discussion of the chosen metrics, the prediction process and the running time comparison. I also find these details to be crucial to this work, especially for rendering the approach more accessible to the non-Bayesian community.
> > >
> > > With these changes and discussions in place, I find that this work will be a good contribution to the community.

---

> > > > ### Author Response · Authors · 2023-08-15
> > > >
> > > > We thank Reviewer kqwq for your additional constructive comments. We are particularly grateful that the reviewer spent time going through our Supplementary Material.
> > > >
> > > > These non-parametric Bayesian methods utilize IBP processes only to regularize the width of latent variables, with fixed network depths. For example, [1] (Chatzis, 2018) and [3] (Singh et al., 2017) proposed to impose IBP prior to regularize the dimensionality of latent variables with constant-depth backbone networks. Specifically, [1] evaluated the proposed approach on two different backbone networks of deep generative models. The distinctive aspect of our contribution is the combined adaptation of both depth and width of VAEs using the beta process and its conjugate Bernoulli process, respectively.
> > > >
> > > > The reason we compared with the “less related” methods (e.g., DVAE [7], CR-VAE [9]) in our paper is because they are the VAE regularization methods that achieve state-of-the-art performance on the benchmark datasets. Our work shows the proposed nonparametric Bayesian structural adaptation method outperforms them.
> > > >
> > > > In the revised version, we will follow Reviewer kqwq’s suggestion and restructure our related work section to elaborate and compare with the more relevant nonparametric Bayesian regularization methods such as [1], [2], [3], [4], and [5] at the same level of detail as what we did in the rebuttal.

---

> > > > ### Author Response · Authors · 2023-08-21
> > > >
> > > > We would like to further elaborate on Chatzis (2018) [1] and beta-Bernoulli VAE (BB-VAE cited and compared in our paper) Singh et al. (2017) [3], both have the same formulation for regularizing latent variable dimensionality using an Indian Buffet Process (IBP). However, their divergence arises in their chosen variational inference techniques: Singh et al. [3] employ a reparametrization trick for gradient computation, while Chatzis [1] utilizes Black Box Variational Inference (BBVI) with control variates. As mentioned before, it's crucial to emphasize that these methods fundamentally differ from our own, which focuses on the adapting width/depth of encoding/decoding network.
> > > >
> > > > In this context, we present a comparison between Chatzis (2018) [3] (referred to as IBP-DGM) and our method, noting the use of the reparametrization trick for variational inference in IBP-DGM.
> > > >
> > > > | Methods | MNIST          | Omniglot        | Caltech    |
> > > > |---------|----------------|-----------------|-----------------|
> > > > | IBP-DGM | 91.55$\pm$0.69 | 124.47$\pm$0.58 | 135.18$\pm$0.83 |
> > > > | Ours    | **82.30$\pm$0.10** | **106.34$\pm$0.01** | **113.54$\pm$0.40** |
> > > >
> > > > We will include the additional results in our revised version.

---

> > > > > ### Comment · Reviewer_kqwq · 2023-08-21
> > > > >
> > > > > I thank the authors for further elaborating on the comparison with the other approaches. I think the differences with the other methods are pretty clear, but comparison was necessary.
> > > > > I have now raised my score for this work.

---

### Official Review · Reviewer_p4pj · 2023-07-05

**Soundness:** 3 good
**Presentation:** 2 fair
**Contribution:** 2 fair
**Rating:** 5
**Confidence:** 4

**Summary:**

The paper proposes a method called AdaVAE that adapt the sizes (depth and width) of the inference and the generative networks in VAEs.
It extends the idea of beta-Bernoulli process VAE to use a beta process over the expected number of activated neurons over depth and Bernoulli processes per layer for the actual activations, followed by a dense Gaussian latent layer.
The paper develops an estimator based on MIWAE to learn AdaVAE.
Results show that AdaVAE can prevents overfitting and lead to performance improvement than previous regularization methods for VAEs.

**Strengths:**

### originality

The proposed AdaVAE for adapting VAE sizes is novel.

### quality

The proposed method is technically sound.

### clarity

The paper did a good presentation of the proposed methods with technical details, which is easy to follow.

### significance

The proposed method is generally applicable to different VAE architecture and is potentially useful as a standard regularization technique.

**Weaknesses:**

### originality

The proposed estimator is a straightforward combination of previous works.

### clarity

The paper is very related to [11,36,54] but only limited discussion is given.
I encourage the author(s) to give some discussions between the AdaVAE and share intuitions of why those previous works (that only adapts the size of the latent layer) are not enough.

Some of the figures with small legends are a bit hard to read.

Figure 4 is purely qualitative.
It's hard to convince the readers that one method yields better representation than others.
It looks like the figure it trying to give an idea on how different clusters are separated.
If it is the case maybe some clustering metrics could be used to assess the quality quantitatively.

### significance

The proposed method can be still computationally heavy to use, compared to a naive one without any structure adaptation.

**Questions:**

In Figure 2, the last few layers of the encoder and the first few layers of the decoder seem to have no neuron activated.
Can the author(s) clarify if it is the case and if so why the VAE still works in this case?

Do we have any computation-wise comparison between the proposed methods and alternative/naive ones?
It is useful to address practical concerns of actually using the method.

**Limitations:**

The paper mentions its limitations on the fixed truncation level and some future work on removing it.

---

> ### Author Rebuttal · Authors · 2023-08-09
>
> We'd like to thank Reviewer p4pj for your constructive comment. They are valuable for our future research.
>
> **Relationship to [11,36, 54]**: Our experiment results in Figure 3 shows the methods as [11,36] only imposing an IBP prior to regularize the dimensionality of latent variables is not sufficient to mitigate overfitting caused by deep encoding/decoding networks. An intuition is large-scale encoding networks tend to learn redundant information or higher levels of feature abstraction from training data and embed it to the latent representation, which causes overfitting problems. Therefore, a more effective way to prevent it is by directly regularizing the network structures, rather than constraining the latent variables. [54] proposes a way to relax the truncation of stick breaking process. A part of our future research is to utilize the technique to improve our method.  We will add this analysis to the camera-ready version.
>
> **Small legends**: We will enlarge all the legends and axis labels in the camera-ready version.
>
> **Quantitative evaluations of Figure 4**: The quantitative evaluations of the VAE variants' performance in Figure 4 are reported in Table 2. The purpose of Figure 4 is only to give readers a qualitative intuition. Moreover, in order to further evaluate the quality of latent representations, we conducted additional experiments and reported downstream classification accuracy in Supplementary Material/Appendix
>  Section 10.3, Table 8. This is a more effective way to assess the clustering.
>
> **Skipping non-activated layers**: We have skip connections as in Eqn. 5, so that we can propagate the last activated layer to the output layer by skipping the non-activated layers in between. We will detail this point in the camera-ready version.
>
> **Computation-wise comparison**: We have computation complexity analysis in Lines 179-182. We also compared the running times between our method and alternative ones and reported and analyzed the results in Supplementary Material/Appendix Section 8.

---

> > ### Comment · Reviewer_p4pj · 2023-08-21
> >
> > Thanks for the response. I'm looking forward to the improvements in the revised version.

---

### Official Review · Reviewer_uhJ9 · 2023-07-06

**Soundness:** 3 good
**Presentation:** 3 good
**Contribution:** 3 good
**Rating:** 5
**Confidence:** 3

**Summary:**

This paper introduces a Bayesian structural adaptation framework that automatically adapts VAE network structures (both encoder and decoder) to the current data.  By modeling the number of hidden layers as a beta process and performing layer-wise dropout regularization with the conjugate Bernoulli process, the proposed model develops a joint ELBO that can optimize all parameters (network structures and latent variables) via the SGD algorithm. Empirical studies are conducted on three visual datasets and two graphical datasets. They also provide extensive ablation studies to show the robustness.

**Strengths:**

(1) The paper addresses one of the common issues in VAE communities. The developed AdaVAE attempts to prevent the overfitting issues by jointly learning the network structures and latent variables under the Bayesian framework. This motivation sounds good, and the novelty of the method is generally ok.

(2) The paper is written clearly and the empirical results show the improvement of the proposed model.

**Weaknesses:**

(1) One of the main concerns comes from the experiment section. The proposed model is only tested on several simple network structures (the number of layer usually less than 25). Additional results on more expressive neural network structures (such as BIVA and NVAE) will improve the quality of this paper.

(2) One of the goal of the proposed model is to prevent the overfitting by adapting the network structures under the Bayesian framework. The authors compared the proposed model with existing VAE regularization methods, and the results show the improvements. Unfortunately, other baselines (such as network structure search methods [1]) are not included in the experiments.

[1] Corinna Cortes et.al. AdaNet: Adaptive Structural Learning of Artificial Neural Networks.



**Questions:**

(1) I suggest that the authors add a training algorithm, which can help the reader understand the algorithm more easily.

(2) From Fig(6) (a) (b), we find that AdaVAE have similar results with cVAE. Can the authors provides deeply analysis?

(3) Given that the proposed AdaVAE aims to find optimal structure for the dataset at hand, it is advisable to include a comparison of network parameters in Table 2.

---

> ### Author Rebuttal · Authors · 2023-08-09
>
> We'd like to thank Reviewer jhJ9 for your constructive comments. They are valuable for our future research.
>
> **Application to expressive VAE variants**: BIVA is an extension of LVAE on which we conducted rigorous tests. We have additional results on BIVA/LVAE on the MNIST dataset as follows:
> | | | |
> |:-:|:-:|:-:|
> | **Methods** |    **-LL** |  **KL**  |
> | BIVA/LVAE      | 116.07$\pm$2.21     | **23.05$\pm$0.15**   |
> | Ours+BIVA/LVAE | **86.07$\pm$0.07**      | 19.78$\pm$0.01  |
>
> We show our framework's performance on NVAE on the cifar10 dataset as below by implementing a simplified NVAE  by designing an encoding and decoding network with a series of NVAE blocks.
>
> | | | |
> |:-:|:-:|:-:|
> | **Methods**   | **Reconstruction loss** | **KL** |
> | NVAE      | **13313$\pm$09**         |   150$\pm$03 |
> | Ours+NVAE | 13600$\pm$30        |   **171$\pm$00** |
>
> We also provided reconstructed samples from the two methods in the attached pdf. These additional results suggest our framework's flexibility on more expressive VAE structures. We will add them to the camera-ready version of our paper.
>
> **Difference with AdaNet**: There are several major differences setting our method and AdaNet apart: 1). AdaNet employs an iterative procedure to search for optimal network structures. Namely, at each iteration, it selects a set of candidate subnetworks and trains/re-trains the expanded alternative models. This is a time-consuming procedure. In contrast, our framework models the growth of VAE network structures as a stochastic process. Consequently, we can jointly infer the network structures and latent variable in a single training pass. 2). AdaNet incrementally constructs feedforward neural network structures in supervised learning settings. It remains uncertain whether it can be effectively extended to VAEs, considering its iterative searching process. Additionally, scalability could be a concern, as AdaNet solely examines network configurations with a maximum of three layers. 3). AdaNet doesn't demonstrate if their method can be applied to different backbone networks, such as CNN and GCN. We will clarify these differences in the camera-ready version of our paper.
>
>
> **Training algorithm**: The pseudocode of our training algorithm was included in the Supplementary Material/Appendix Section 4. We conducted additional experiments to analyze its convergence in training in Appendix Section 6. We have also included our codes in the Supplementary Material.
>
> **Similar results between AdaVAE and cVAE in Figure 6**: The similar results between cVAE and AdaVAE for a smaller number of layers in Figure 6 (a) and (b) are due to the convolutional layers' preventive effect to overfitting. In comparison, Figure 3 shows that fully connected networks are more sensitive to overfitting. However, for deeper network structures, Figure 6 shows that cVAE still suffers from overfitting. AdaVAE, in contrast, can successfully mitigate it. We will elaborate on the results in the camera-ready version.
>
> **Comparison of network parameters**: First, we reported detailed parameter settings of the baseline methods and ours in Supplementary Material/Appendix Section 5.3. Second, since the baselines activate the whole network structures, so the total number of parameters for encoding and decoding network are $2\times O\times O
> \times L$, where $O$ denotes the maximum number of neurons per layer (i.e., width) and $L$ is the number of layers. For layers $L=25$ and width of $O=200$, they have $2M$ parameters. With the same size of truncation, AdaVAE only activates a part of it in general and fits the activated structures to data. Thus, the activated number of parameters (neuron activation percentage) for $T=L=25$ are as follows:
>
> | Methods |   MNIST      | Omniglot     | Caltech101      |
> |---------|--------------|--------------|-----------------|
> | Ours    | 0.32M(16%) | 0.36M(18%) | 0.21M (10.5%) |
>
> As the table shows AdaVAE uses a smaller number of parameters to achieve state-of-the-art performance. We will include the results in the camera-ready version of our paper.

---

> > ### Comment · Reviewer_uhJ9 · 2023-08-15
> >
> > I thank the authors for their clarifications, which address most of my concerns. I would like to see this paper at the conference.

---

> > > ### Author Response · Authors · 2023-08-15
> > >
> > > We are truly grateful for Reviewer uhj9's positive feedback and endorsement of our research. We kindly inquire whether the reviewer might consider revising our rating, which could potentially enhance our prospects of showcasing our work at the upcoming conference. Your consideration would be greatly valued.

---

### Official Review · Reviewer_9dFM · 2023-07-06

**Soundness:** 3 good
**Presentation:** 3 good
**Contribution:** 3 good
**Rating:** 7
**Confidence:** 2

**Summary:**

The paper proposes a novel VAE structural adaptation strategy called AdaVAE based on Bayesian model selection to enhance model performance. It introduces a scalable estimator that facilitates joint inference on both encoding/decoding network structures and latent variables. The paper conducts a comprehensive analysis of AdaVAE's regularization capabilities and demonstrates its ability to effectively mitigate overfitting in both shallow and deep VAE models and achieve state-of-the-art performance. The versatility of AdaVAE is showcased by demonstrating its compatibility with different types of VAE backbone networks. It can also be readily applied to various VAE variants, thereby enhancing their performance.

The main contributions are:
- Proposes AdaVAE, a novel VAE structural adaptation strategy based on Bayesian model selection to enhance model performance.
- Introduces a scalable estimator that facilitates joint inference on both encoding/decoding network structures and latent variables.
- Conducts a comprehensive analysis of AdaVAE's regularization capabilities and demonstrates its ability to effectively mitigate overfitting in both shallow and deep VAE models and achieve state-of-the-art performance.
- Showcases the versatility of AdaVAE by demonstrating its compatibility with different types of VAE backbone networks.
- Can be readily applied to various VAE variants, thereby enhancing their performance.

**Strengths:**

1. The authors introduce a novel Variational Autoencoder (VAE) structural adaptation strategy, dubbed AdaVAE, which employs Bayesian model selection as a mechanism to enhance model performance. This innovative approach pushes the boundaries of current practices in the field and sets a precedent for future explorations.

2. The study further contributes by proposing a scalable estimator. This facilitates joint inference on not only the structures of the encoding and decoding networks but also the latent variables. This dual focus enhances the model's applicability and comprehensiveness, potentially opening new avenues in inferential methodologies.

3.A thorough analysis is presented on AdaVAE's regularization capabilities, showcasing its efficacy in mitigating overfitting across both shallow and deep VAE models. This is a critical point, as it demonstrates the proposed method's capability to achieve state-of-the-art performance across varying levels of complexity.

4. The versatility of AdaVAE is effectively demonstrated, as the authors show its compatibility with different types of VAE backbone networks. This level of adaptability reinforces the model's potential for broad application and usability within the field.
The authors execute a rigorous evaluation of the proposed method on three benchmark datasets: MNIST, Omniglot, and Caltech101 Silhouettes. This broad evaluation allows a comprehensive understanding of AdaVAE's performance and robustly positions it in relation to other state-of-the-art methods.

5. The paper includes a qualitative evaluation of the latent representations, an element that strengthens the argument for the proposed method's compatibility with VAE regularization methods. This kind of analysis enhances the empirical rigor of the study and provides an additional perspective on the utility of AdaVAE.

6. The manuscript offers a theoretical analysis of the proposed method and derives a tight lower bound with a high signal-to-noise ratio for parameter gradients. This theoretical grounding ensures the model is not only empirically valid but also theoretically sound.

7.The manuscript is particularly well-executed in terms of its structure and style. The clear writing, logical organization, and the authors' ability to elucidate complex concepts makes it accessible and easy for readers to grasp the proposed method and its evaluation.

In essence, the paper is a substantial contribution to the field, demonstrating methodological innovation, theoretical robustness, and a thorough evaluation, all of which underscore the potential and value of AdaVAE within the realm of VAE structural adaptation.

**Weaknesses:**

This manuscript is laudable for its comprehensive coverage, including a lucid motivation, rigorous theoretical proofs, and a thorough comparative analysis with benchmark models across various datasets. It demonstrates a high level of academic rigor and makes a compelling case for the proposed model.

However, I do have a minor suggestion to enhance the paper further. While the evaluation conducted on Caltech101, among other datasets, certainly contributes to the robustness of the results, I believe there could be value in extending this evaluation to include more real-life, challenging datasets. Such datasets, replete with their inherent complexities, could serve to more thoroughly test and validate the model. This, in turn, would potentially bring the proposed model closer in comparison to other state-of-the-art generative models. It would also ensure that the model's performance is tested in conditions that mirror real-world applications, reinforcing the practical relevance and applicability of the model.

**Questions:**

Please refer to the above sections for detailed discussion.


**Limitations:**

Yes, there is a section at the end of the paper discussing limitations and future research opportunities.

---

> ### Author Rebuttal · Authors · 2023-08-09
>
> We'd like to thank Reviewer 9dFM for your time and your constructive comments. They are valuable for our future research.
>
> We agree with Reviewer 9dFM that extending the evaluation to include more real-life, challenging datasets will be valuable. It is part of our future work. In addition, we conducted some additional experiments to test our framework on CIFAR-10. We implemented a simplified NVAE by designing an encoding and decoding network with a series of NVAE blocks. We report the performance as below:
>
> | | | |
> |:-:|:-:|:-:|
> | **Methods**   | **Reconstruction loss** | **KL** |
> | NVAE      | **13313$\pm$09**       |  150$\pm$03 |
> | Ours+NVAE | 13600$\pm$30    | **171$\pm$00** |
>
> We also provided reconstructed samples from the two methods in the attached pdf.
>
> We will add more additional results on real-life datasets in the camera-ready version of our paper.

---

> > ### Comment · Reviewer_9dFM · 2023-08-21
> >
> > Thanks for the rebuttal. With everything considered, I'd stay with my current rating.

---

### Author Rebuttal · Authors · 2023-08-09

Attached here is the reconstructed samples of the cifar-10 dataset (related to the additional experimental results.)

---

### Decision · Program_Chairs · 2023-09-21

**Decision:**

Accept (poster)

**Comment:**

This is definitely a nice and clear paper. The authors worked hard to convince the reviewers about the novelty compared to existing work and they succeeded. Acceptance is therefore recommended.